# FASG: Feature Aggregation Self-training GCN for Semi-supervised Node Classification

## Abstract

Recently, *graph convolutioal networks* (GCNs) have achieved significant success in many graph-based learning tasks, especially for node classification, due to its excellent ability in representation learning. Nevertheless, it remains challenging for GCN models to obtain satisfying predictions on graphs where few nodes are with known labels. In this paper, we propose a novel self-training algorithm based on GCN to boost semi-supervised node classification on graphs with little supervised information. Inspired by self-supervision strategy, the proposed method introduces an ingenious checking part to add new nodes as supervision after each training epoch to enhance node prediction. In particular, the embedded checking part is designed based on aggregated features, which is more accurate than previous methods and boosts node classification significantly. The proposed algorithm is validated on three public benchmarks in comparison with several state-of-the-art baseline algorithms, and the results illustrate its excellent performance.

## 1 Introduction

*Graph convolutional network* (GCN) can be seen as the migration of *convolutional neural network* (CNN) on non-Euclidean structure data. Due to its its excellent ability in representation learning, GCN has achieved significant success in many graph-based learning tasks, including node clustering, graph classification and link prediction (Dwivedi et al., 2020). Kipf & Welling (2016) proposed a GCN mode from the perspective of spectrogram theory and validated its effectiveness on semi-supervised node classification task. Subsequent models such as GraphSAGE (Hamilton et al., 2017), GAT (Veličković et al., 2017), SGCN (Wu et al., 2019) and APPNP (Klicpera et al., 2018) designed more sophisticated neighborhood aggregation functions from spatial or spectral views. These methods obtain much more effective results on semi-supervised node classification than traditional methods such as MLP, DeepWalk (Perozzi et al., 2014), etc. However, the prediction accuracy of such GCN models depends largely on the quantity and quality of supervised information, and it will decrease significantly when the quantity of labeled nodes is quite small (Li et al., 2018). The main reason lies that scarce supervised information is difficult to be spread far away in the graph so that unlabeled nodes are hardly to make full use of supervised information for prediction.

Addressing the above issue, many studies have been devoted to improving the representation ability by designing multi-layer GCN model (Li et al., 2019). However, the representation ability of GCN, as illustrated in Kipf & Welling (2016), can hardly be improved by simply stacking layers just like MLP. Moreover, stacking too many layers tends to cause over-smoothing (Xu et al., 2018) that makes all node embeddings indistinguishable. Alternatively, Li et al. (2018) proposed to improve the reasoning ability of GCN models by applying self-training techniques on the training. Rather than trying to enhance the expressive ability of the model, the self-training strategy prefers to expand the supervised information by adding unlabeled nodes with high confidences to the training set at each round. Following this line, Sun et al. (2019) proposed a *multi-stage self-training strategy* (M3S) to enrich the training set, which uses deep cluster (Caron et al., 2018) and an aligning mechanism to generate pseudo-labels of nodes for updating of the training set. Later, Zhou et al. (2019) proposed a dynamic self-training framework to continuously refresh the training set by directly using the output of GCN without a checking part. In general these self-training algorithms generate pseudo-labels using relatively simple checking mechanism, which may introduce false labels as supervision information and prevent the improvement of prediction accuracy.

In this paper, we propose a novel *feature aggregation self-training GCN* (FASG) algorithm for semi-supervised node classification. We firstly propose a lightweight classifier that applies linear SVM on aggregated node features, and validate that it achieves comparable performance to popular GCN approaches. Furthermore, this classifier is served as a checking part in the multi-round training process to generate pseudo-labels, which are used to filter unreliable nodes when expanding the supervised information. By fully considering the structural information of graph nodes, the newly developed checking part is able to improve the accuracy of the generated pseudo-labels and finally boost the node classification. Finally, we illustrate that the proposed self-training strategy can be integrated with various existing GCN models to improve the prediction performance.

The proposed algorithms is validated in comparison with several state-of-the-art baseline algorithms in three public benchmarks, and the experimental results illustrate that the proposed algorithm outperforms all compared algorithms in general on all benchmarks. We will release the source code upon publication of this paper.

## 2 RELATED WORK

In the past decade CNN has achieved great success in many areas of machine learning (Krizhevsky et al., 2012; LeCun et al., 1998; Sermanet et al., 2012), but its applications are mainly restricted in dealing with Euclidean structure data (Bruna et al., 2013). Consequently, in recent years more and more studies are devoted to learning the representation on non-Euclidean structure data such as graph.

*Graph neural network* (GNN) plays an important role in the field of graph representation learning, which can learn the representation of nodes or the whole graph. There are many famous GNN architectures including GCN (Kipf & Welling, 2016), graph recurrent neural network (Hajiramezanali et al., 2019) and graph autoencoder (Pan et al., 2018). As one of the most important architecture of GNN, GCN can be roughly categorized into spectral and spatial approaches. The spectral approaches (Bruna et al., 2013) define convolution operation by Laplacian feature decomposition of the graph, thereby filtering the graph structure in the spectral domain. On the basis of the Chebyshev polynomial (Defferrard et al., 2016) of the graph Laplacian matrix, Kipf & Welling (2016) proposed a much simper GCN framework that limits the filter to the first-order neighbor around each node. On the other hand, spatial approaches implement convolution in spatial domain by defining aggregation functions and transform functions. Notable work includes GraphSAGE (Hamilton et al., 2017) that transformed representation learning into a formal pattern called aggregation and combination and proposed several effective aggregation strategies such as mean-aggregator and max-aggregator, and GAT (Veličković et al., 2017) that focuses on the diversity in connected nodes and leverages self-attention mechanism to learn the important information in neighborhoods. Although these models have achieved far better performance on node classification than traditional methods, they still suffer from scarce supervised information due to the limitation on GCN layers making it hard to transform the supervised information to the entire graph.

Self-training is an ancient and classic topic in the NLP field before deep learning era (Hearst, 1991; Riloff et al., 1999; Rosenberg et al., 2005; Van Asch & Daelemans, 2016), and has recently been introduced into semi-supervised node classification. For making full use of supervised information to improve the prediction accuracy, Li et al. (2018) proposed to improve GCN model by self-training mechanism, which trains and applies a base model in rounds, and adds nodes with high confidences as supervision after each round. The newly added nodes are expect to be beneficial to predict rest nodes so as to enhance the final performance of the model. Following this line, the M3S training algorithm Sun et al. (2019) pretrains a model over the labeled data, and then assigns pseudo-labels to highly confident unlabeled samples that are considered as labeled data for the next round of the training. Later, Zhou et al. (2019) proposed a dynamic self-training GCN that generalizes and simplifies previous by directly using the output of GCN without a checking part to continuously refresh the training set. Similarly, Yang et al. (2020) proposed *self-enhanced GNN* (SEG) to improve the quality of the input data using the outputs of existing GNN models. These self-training methods expand the labeled node set with relatively simple checking mechanism or even directly using the output of GCN, as a result they may introduce noise as supervision and thus hurt the final prediction performance.

## 3 PRELIMINARIES

An attributed relational graph of $n$ nodes can be represented by $G = (V, E, X)$, where $V = \{v_1, v_2, ..., v_n\}$ denotes the set of $n$ nodes, and $E = \{e_{ij}\}$ is the edge set. $X = \{x_1, x_2, ...x_n\} \in R^{n \times d}$ is the set of attributes of all nodes, where $x_i$ is the $d$-dimensional attribute vector associated with node $v_i$. Adjacency matrix $A = \{a_{ij}\} \in R^{n \times n}$ denotes the topological structure of graph $G$, where $a_{ij} > 0$ if there is an edge $e_{ij}$ between node $v_i$ and $v_j$ and $a_{ij} = 0$ otherwise.

For semi-supervised node classification, the node set $V$ can be split into a labeled node set $V_L \in V$ with attributes $X_L \in X$ and an unlabeled one $V_U = V \backslash V_L$ with attributes $X_U = X \backslash X_L$. We assume each node belongs to exactly one class, and denote $Y_L = \{y_i\}$ the ground-truth labels of node set $V_L$ where $y_i$ is the class label of node $v_i \in V_L$.

The aim of semi-supervised node classification is to learn a classifier from the graph and known node labels $Y_L$, and use it to predict labels for unlabeled nodes $V_U$. Define a classifier $f_\theta : (\tilde{Y}_L, \tilde{Y}_U) \leftarrow f_\theta(X, A, Y_L)$, where $\theta$ denotes the parameters of model, $\tilde{Y}_L$ and $\tilde{Y}_U$ are the predicted labels of nodes $V_L$ and $V_U$ respectively. In general, we want the predict labels $\tilde{Y}_L$ is close to the ground-truth labels $Y_L$ as possible in favor of

$$\theta^* = \arg\min_\theta d(\tilde{Y}_L, Y_L) = \arg\min_\theta d(f_\theta(X, A, Y_L), Y_L), \tag{1}$$

where $d(\cdot, \cdot)$ is a distance measure between two label sets.

In recent years GCN has become a popular model for semi-supervised node classification, which aggregates a structural feature for each node and use the formed features, rather than the initial attributes $X$, for label prediction.

## 4 THE PROPOSED METHOD

In this section, we will elaborate our proposed framework, namely *feature aggregation self-training GCN* (FASG), for semi-supervised node classification. Firstly, we do analysis of pseudo-labels to explain the importance of checking part in self-training framework. Secondly, we illustrate the design of checking part in our framework and show the superiority of our checking part in graph networks. Then we elaborate every part of the framework and display the FASG training algorithm. Finally we integrate our framework with various GNN models.

### 4.1 ANALYSIS OF PSEUDO-LABELS

It is common for existing self-training GCN models to assign pseudo-labels to highly confident nodes and expand them as supervised information. Therefore, the quality of the generated pseudo-labels is crucial for node classification and the wrongly introduced supervision information may hurt the final prediction performance. Table 1 summarizes the prediction accuracy of the GCN model (Kipf & Welling, 2016) on Cora when it is trained given different ratios of falsely labeled nodes. It is shown that the accuracy decreases significantly with the ratio of bad training nodes increasing.

Table 1: Performance of GCN model trained with different ratio of bad training nodes.

| Ratio | 0% | 10% | 20% | 30% | 40% | 50% | 60% |
|---|---|---|---|---|---|---|---|
| Acc | 81.9% | 77.9% | 71.5% | 69.1% | 65.4% | 57.9% | 56.3% |

### 4.2 CHECKING PART WITH FEATURE AGGREGATION

To guarantee the quality of the generated pseudo-labels, we develop a delicate checking part in the assistance of feature aggregation. The implementation of feature aggregation can be described as $X^{aggre} = \tilde{D}^{-1}\tilde{A}X$, where $D$ is digree matrix of the graph, $\tilde{D} = D + I$, $\tilde{A} = A + I$. We use deep graph library DGL (Wang et al., 2019) to implement feature aggregation.

For illustration of the effectiveness of feature aggregation, we apply t-SNE (Maaten & Hinton, 2008) to visualize the aggregated features of each node on the Cora dataset in Fig 1, where $feat_i$ denotes the features aggregated from the original features for $i$ times. As shown in Fig. 1(a), the original node features are mixed together and are difficult to distinguish. As the fusion of node features going deeper from $feat_1$ to $feat_4$, nodes with the same label tend to aggregate into clusters in 2-D space. However, the cluster boundaries become blur again after the aggregation goes up to a certain level, e.g.. $feat_{15}$ and $feat_{20}$.

Furthermore, we apply linear svm (Cortes & Vapnik, 1995) on aggregated features $feat_5$ to form a classifier, and report its performance in Table 2 in comparison with several GCN models on three citation networks, Cora, CiteSeer and PubMed. Clearly, this relatively simple classifier is able to achieve comparable performance with popular GCN models due to the representation ability of aggregated features.

As for the self-training mechanism, we employ the above classifier that combines feature aggregation with line svm to serve as check part for generation of pseudo-labels of nodes. In Fig. 2, we compare the quality of pseudo-labels generated by different checking mechanisms including plain self-training method (Li et al., 2018), deep cluster Sun et al. (2019) and the proposed checking part with feature aggregation. It is shown that our method introduces less bad training nodes than the compared methods in different label rates on both Cora and CiteSeer, which accounts for the better performance on node classification shown in Sec. 5.

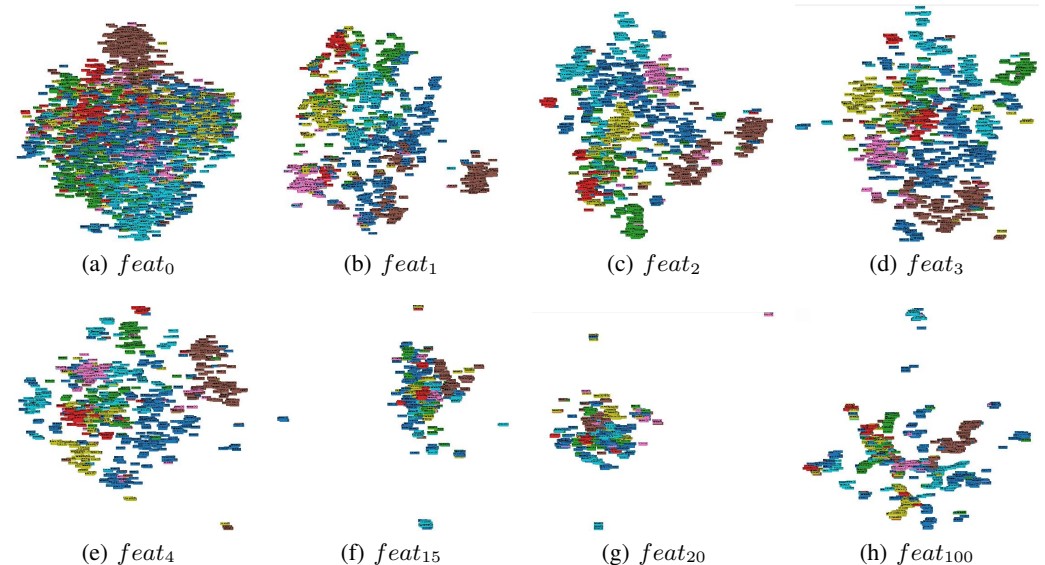

| (a) $feat_0$ | (b) $feat_1$ | (c) $feat_2$ | (d) $feat_3$ |
| (e) $feat_4$ | (f) $feat_{15}$ | (g) $feat_{20}$ | (h) $feat_{100}$ |

Figure 1: T-SNE visualization of aggregated features in the Cora dataset.

Table 2: Performance of aggregation and classic methods

| Dataset | MLP | ChebyNet | GCN | GAT | feat5+svm |
|---------|------|----------|-------|-------|-----------|
| Cora | 55.1% | 81.2% | 81.5% | 83.0% | 83.2% |
| CiteSeer | 46.5% | 69.8% | 70.3% | 72.5% | 72.3% |
| PubMed | 71.4% | 74.4% | 79.0% | 79.0% | 78.8% |

### 4.3 MULTI-STAGE SELF-TRAINING FRAMEWORK

The overall framework of the proposed *feature aggregation self-training GCN* (FASG) algorithm is illustrated in Fig. 3. Instead of using deep cluster and aligning mechanism, we firstly apply feature aggregation and linear SVM classifier to build a checking part. After each training round we use

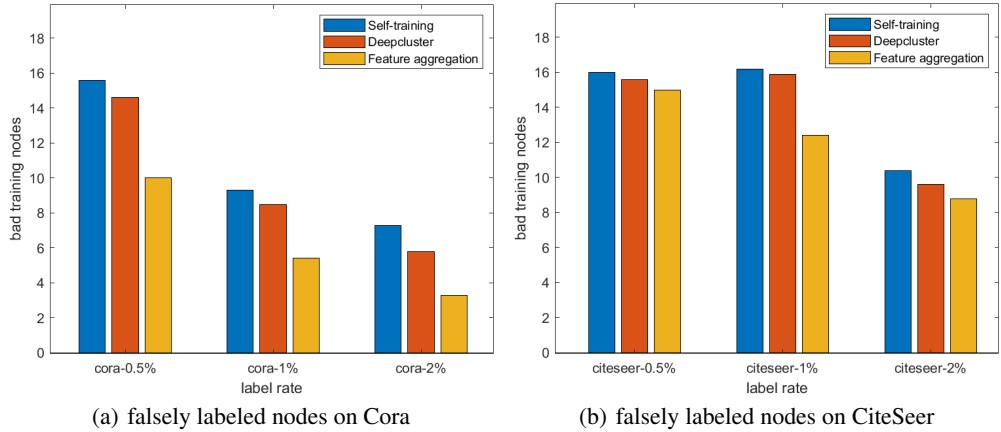

(a) falsely labeled nodes on Cora        (b) falsely labeled nodes on CiteSeer

Figure 2: Comparison of the number of falsely labeled nodes introduced by different checking mechanism.

both the output GCN confidence and the checking part to choose reliable nodes as supervised ones at the next round. The training iterates $K$ rounds and then output the final predictions of unlabeled nodes.

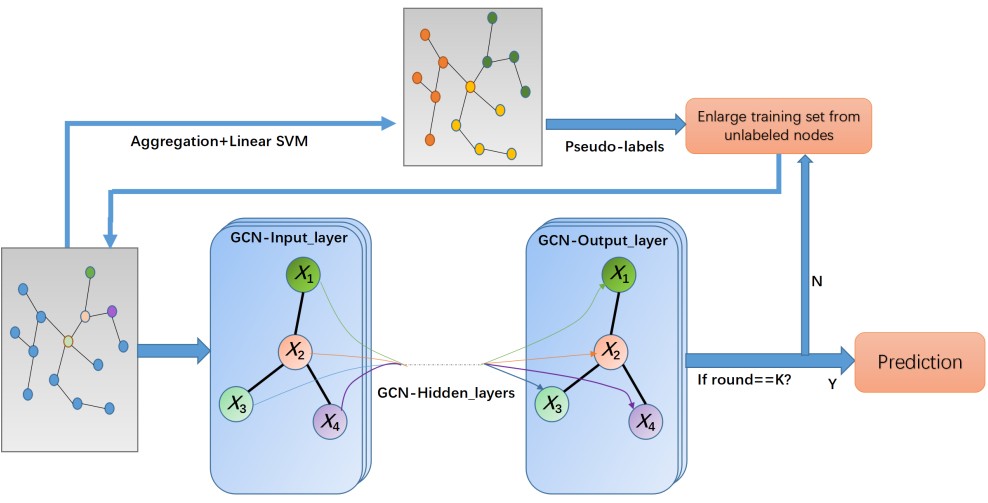

Figure 3: The overall FASG framework for semi-supervised node classification.

The proposed FASG algorithm is described in details in Algo. 1. At the beginning, we concatenate features from $feat_0$ to $feat_{10}$ and put them into a linear SVM to build the checking part. At each round if the output of a node predicted by the GCN model is consistent with its pseudo-label generated by the checking part, then we tend to expand this node with high certainty to the supervised set. To avoid expanding too much nodes at one time, only $t$ nodes with top confidence are checked at each round. Note that the base GCN model in Algo. 1 is not specified, i.e. the proposed FASG algorithms can be integrated with various GCN models to boost node classification, of which the effectiveness is validated in Table. 6.

---

**Algorithm 1** The FASG Algorithm

---

**Input:**
    $G = (V, E, X)$: the input graph.
    $A$: the adjacent matrix of graph $G$.
    $L, U$: the labeled and unlabeled node set of respectively.
    $GCNconv(\cdot)$: the base GCN model.
    $SVM(\cdot)$: the linear SVM classifier.
    $K$: the number of self-training rounds.
**Output:**
    predictions of all the unlabeled nodes $\tilde{Y}_U$;
1: Conduct feature aggregation to generate $feat_1, \ldots, feat_9$.
2: Form concatenation $feat \leftarrow [feat_1, \ldots, feat_9]$.
3: Generate pseudo-labels $Y'_U = SVM(feat, L, U)$.
4: Let $L' \leftarrow L, U' \leftarrow= U$;
5: **for** $k = 1$ to $K$ **do**
6:     Train GCN model and get predictions and confidence matix:
    $\tilde{Y}_U, M = GCNconv(A, X, L', U')$.
7:     **for** each class j **do**
8:         Select $t$ nodes $\{v_{j1}, \ldots, v_{jt}\}$ in $U'$ with top confidences.
9:         **for** $i = 1$ to $t$ **do**
10:             **if** $\tilde{y}_{ji}$ equals $y'_{ji}$ **then**
11:                $L' \leftarrow L' \cup \{v_{ji}\}, U' \leftarrow U' \backslash \{v_{ji}\}$.
12:             **end if**
13:         **end for**
14:     **end for**
15: **end for**
16: Compute the final predictions $\tilde{Y}_U = GCNconv(A, X, L', U')$.
17: **return** $\tilde{Y}_U$

---

## 5 EXPERIMENTS

### 5.1 EXPERIMENTAL SETTINGS

We conduct experiments on three open graph datasets derived from citation networks (including Cora, CiteSeer, PubMed) (Sen et al., 2008) for the semi-supervised node classification task. In these citation networks nodes denote documents whose features are formed by bag-of-words representations, and edges denote their relationships with labels indicating what field the corresponding documents belong to.

Though our framework can be integrated with various GNN models, we choose plain GCN (Kipf & Welling, 2016) as the base model in Algo. 1 in this experiment. Specifically, we set the number of GCN layers n_layers=2, learning rate lr=1e-2, training epochs=600, weight_decay=5e-4 for the GCN model, and fix $t = 10$ in Algo. 1. Similar to the M3S algorithm (Sun et al., 2019), we also regard the option of rounds $K$ as a hyper-parameter and assign the most suitable $K$ for each testing of different label rate. We choose $K$ as 40,10,5,4,4 for Cora dataset, 30,25,15,10,10 for CiteSeer and 5,4,3 for PubMed. The label rate indicates the amount of labeled nodes, which are randomly chosen from the whole node set under an extra measures that is to guarantee the balance between different classes. For each trial we repeat the testing 10 times and report the mean accuracy.

### 5.2 COMPARISON WITH BASELINE ALGORITHMS

The compared baseline algorithms in this experiment include traditional learning method such as Node2Vec (Grover & Leskovec, 2016) ,LP (Wu et al., 2012) and classic GNN approach such as GCN (Kipf & Welling, 2016), GAT (Veličković et al., 2017) and ChebNet (Defferrard et al., 2016). We also include Co-training and Self-training proposed by Li et al. (2018) and other self-learning based approaches MultiStage (Sun et al., 2019), M3S (Sun et al., 2019) as baseline. The relevant experimental settings and results are all taken from original papers.

Table 3: Comparison of prediction accuracy on Cora dataset

| Label Rate | 0.5% | 1% | 2% | 3% | 4% |
|---|---|---|---|---|---|
| Node2Vec | 32.4% | 44.4% | 50.2% | 54.5% | 57.4% |
| LP | 57.6% | 61.0% | 63.5% | 64.3% | 65.7% |
| Cheby | 38.0% | 52.0% | 62.4% | 70.8% | 74.1% |
| GCN | 50.6% | 58.4% | 70.0% | 75.7% | 76.5% |
| GAT | 49.0% | 60.7% | 72.8% | 77.7% | 79.7% |
| Co-training | 53.9% | 57.0% | 69.7% | 74.8% | 75.6% |
| Self-training | 56.8% | 60.4% | 71.7% | 76.8% | 77.7% |
| MultiStage | 61.1% | 63.7% | 74.4% | 76.1% | 77.2% |
| M3S | 61.5% | 67.2% | 75.6% | 77.8% | 78.0% |
| FASG | **62.8%** | **68.5%** | **76.1%** | **78.0%** | **80.3%** |

Table 4: Comparison of prediction accuracy on CiteSeer dataset

| Label Rate | 0.5% | 1% | 2% | 3% | 4% |
|---|---|---|---|---|---|
| Node2Vec | 24.9% | 29.1% | 34.4% | 35.7% | 38.8% |
| LP | 37.7% | 41.6% | 41.9% | 44.4% | 44.8% |
| Cheby | 31.7% | 42.8% | 59.9% | 66.2% | 68.3% |
| GCN | 44.8% | 54.7% | 61.2% | 67.0% | 69.0% |
| GAT | 44.6% | 53.7% | 64.6% | 66.4% | 69.3% |
| Co-training | 42.0% | 50.0% | 58.3% | 64.7% | 65.3% |
| Self-training | 51.4% | 57.1% | 64.1% | 67.8% | 68.8% |
| MultiStage | 53.0% | 57.8% | 63.8% | 68.0% | 69.0% |
| M3S | 56.1% | 62.1% | 66.4% | 70.3% | 70.5% |
| FASG | **58.3%** | **66.4%** | **70.2%** | **70.3%** | **70.8%** |

The comparison of these algorithms on the three benchmarks is summarized in Tables 3, 4 and 5 respectively. It is observed that GNN-based approaches surpass traditional learning approaches in general on all three datasets. By adopting multi-rounds training strategy and expanding the supervised information iteratively, the algorithms based on self-training mechanism achieve remarkable improvement in prediction accuracy, especially when the label rate is quite small. Furthermore, the proposed FASG algorithm outperforms all baseline algorithms in all tested scenarios. The superiority of our method derives from the delicate checking part based feature aggregation, which is able to guarantee the high quality of the expanded supervised information as illustrated in Fig. 2.

## 5.3 ABLATION STUDIES

### 5.3.1 THE NUMBER OF TRAINING ROUNDS

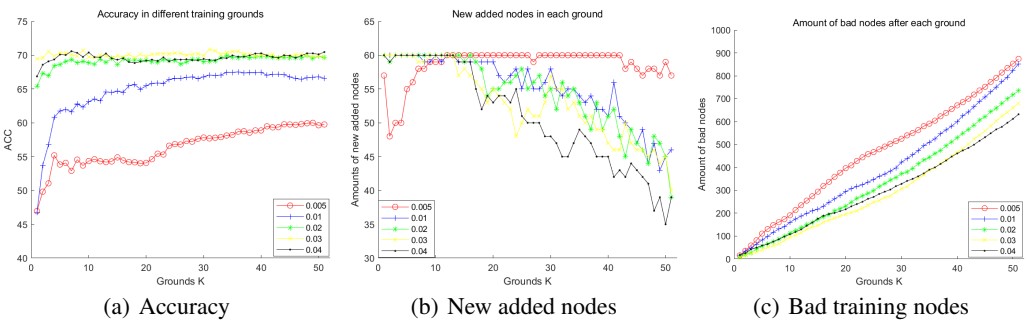

| (a) Accuracy | (b) New added nodes | (c) Bad training nodes |
|---|---|---|

Figure 4: Accuracy, new added nodes and bad training nodes in each round

Table 5: Comparison of prediction accuracy on PubMed dataset

| Label Rate | 0.03% | 0.05% | 0.1% |
|---|---|---|---|
| Node2Vec | 37.2% | 38.2% | 42.9% |
| LP | 58.3% | 61.3% | 63.8% |
| Cheby | 40.4% | 47.3% | 51.2% |
| GCN | 51.1% | 58.0% | 67.5% |
| GAT | 50.6% | 59.1% | 65.0% |
| Co-training | 55.5% | 61.6% | 67.8% |
| Self-training | 56.3% | 63.6% | 70.0% |
| MultiStage | 57.4% | 64.3% | 70.2% |
| M3S | 59.2% | 64.4% | 70.6% |
| FASC | **60.1%** | **65.2%** | **70.7%** |

In order to reveal how our algorithm is affected by the number of training rounds $K$, we report the numbers of newly added nodes, bad training nodes and the prediction accuracy on the CiteSeer dataset for different label rates with increasing $K$ from 0 to 50. Note that when $K$ is 0, the framework degrades to the plain GCN model. As shown in Fig. 4(a), accuracies grow rapidly during the first few rounds for all label rates. For a small label rate (e.g. 0.005), the accuracy tends to grow continuously with $K$ increasing. On the contrary, for a relatively large label rate (e.g. 0.04) the accuracy will reach the peak rapidly with a small $K$ and saturate afterward.

Fig. 4(b) shows the number of newly added nodes after each training round, which is consistent with the change of the accuracy in Fig. 4(a). There are numbers of newly nodes that are expanded as supervision information at each round for a small label rate, so the accuracy is improved continuously. While, for a relatively large label rate, the number of newly added nodes drops markedly after a few training rounds, which results in the saturation of the accuracy.

Table 6: Performance of our framework integrated with different base GNN models

| GNN Model | GCN | GAT | APPNP | GS-M | GS-P |
|---|---|---|---|---|---|
| Acc of normal GNN | 44.8% | 46.8% | 45.7% | 40.9% | 34.9% |
| Acc in our framework | 58.3% | 59.8% | 57.6% | 59.1% | 51.0% |

### 5.3.2 Integration with different base GCN models

As described in Sec. 4.3, the proposed FASG algorithm can be integrated with various base GCN models to improve their prediction performances. For validation, we combine FASG with several popular GCN models, and report their prediction accuracy on the CiteSeer dataset with label rate 0.5% in Table 6, where GS-M and GS-P represent GraphSage with mean and maxpool aggregator respectively. It is shown that all tested base GCN models achieve similar performances, and they are all benefitted significantly in prediction accuracy by applying our FASG to expand supervised information iteratively.

## 6 Conclusion

In this paper, we firstly analyzed the limitations of plain GCN models in dealing with semi-supervised node classification tasks, and subsequently proposed a feature aggregation self-training GCN algorithm aiming to improve the prediction accuracy. Our algorithm iteratively expand reliable nodes into the supervised set by checking both the GCN outputs and the pseudo-labels of nodes that are generated through applying a linear SVM classifier on the aggregated features. This checking mechanism is able to provide supervised information of better quality than previous methods and boosts the final node classification significantly. In experiments, the proposed algorithm outperforms state-of-the-art baseline algorithms in general on all tested benchmarks, especially when the ratio of labeled nodes is quite small.

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
