# OpenReview forum: "FASG: Feature Aggregation Self-training GCN for Semi-supervised Node Classification"
_ICLR.cc/2021/Conference — Reject_

### Official Review · AnonReviewer2 · 2020-10-26
**my review**

**Rating:** 3
**Confidence:** 5

**Review:**

This paper presents a self-training algorithm based on GCN to improve the semi-supervised node classification on graphs. The key idea is to add new nodes with high confidence as supervision to enlarge the labeled nodes. Although the experimental results show the proposed method outperforms or performs similarly to baseline methods, the paper has several weaknesses. First the presented approach is not clearly introduced, with inconsistent statements on building the checking part, and lack of details on how to calculate the confidence to add the new nodes. Second, the novelty of the presented approach is limited, as adding unlabeled samples with high confidence is not a novel idea. Third, the paper writing should be improved, as there are errors.

In Table 2, GAT has better or similar performance comparing to feat5+SVM. Why not using GAT feature aggregation for building the checking part? And, in page 5, it says “At the beginning, we concatenate features from feat0 to feat10 and put them into a linear SVM to build the checking part. ”.  So, it is confusing how the checking part is built. In Algorithm 1,  feat1 to feat9 were concatenated in line 2. The statements are not consistent in the whole paper.

In the line 6 of Algorithm 1, Train GCN model and get predictions and confidence matix:   How the confidence is calculated?

There are writing errors to correct, such as   Due to its its excellent,   confidence matix:

---

### Official Review · AnonReviewer4 · 2020-10-28
**Review comments for FASG**

**Rating:** 4
**Confidence:** 4

**Review:**

The paper proposes an algorithm combining SVM and GCN to solve the node classification problem in label-less scenarios. The proposed model uses a self-training mechanism to generate labels and features and integrates SVM to improve the confidence level of the labels.

Strong points:
-It first uses relationship information among nodes for training an SVM. The algorithm based on the SVM-checking mechanism achieves remarkable improvement in prediction accuracy, especially when the label rate is quite small.

Weak points:
- The description in the SVM classifier experiment section is not clear and detailed enough. How does it deal with different multi-class classification problems?   Perhaps, it should verify whether updating the SVM with pseudo-labels can improve the performance of the model in the case of low labeling rates.

- According to the analysis in the paper, GAT and SVM have similar performance. Why did the authors choose to use SVM as the checking part rather than GAT? Some experiments may be required to show the advantages of using SVM checking.

- The proposed method should be compared with some previous SOTA methods (e.g., Union and Intersection method proposed in "Deeper Insights into Graph Convolutional Networks for Semi-Supervised Learning." (2018).)

- The proposed method is not innovative enough, because it just employs a different check algorithm.

---

### Official Review · AnonReviewer3 · 2020-10-28
**Well motivated self-training based semi-supervised framework for node classification**

**Rating:** 4
**Confidence:** 4

**Review:**

### Summary
This paper proposes a self-training based semi-supervised framework for node classification using Graph Neural Networks when the amount of labelled data is very limited. Self-training is performed by incorporating highly confident samples with their corresponding predicted class as the pseudo label. Authors show that incorporation of correct pseudo labels is a crucial step as the performance degrades rapidly with the incorporation of wrong labels. This work ensures high quality of pseudo labels by a "checking part" with feature aggregation. Aggregated features with linear SVM performs comparably with GNN methods.

### Strong Points
 1. Well motivated paper with good performance
 2. Proposed approach uses self-training based approach with linear SVM which performs well when the amount of labelled data is scarce.
 3. The framework can be easily incorporated with any GNN based approaches.

### Weak Points
 1. I am little surprised that the feature aggregation performs comparably to GCN considering the fact that it is essentially the forward pass of GCN. The improvement could be influenced more because of incorporating the extra examples using self-training than feature aggregation.
 2. Generally easy (very similar to training distribution) examples end up with high confidence scores. Incorporating such nodes might make hard nodes (closer to decision boundary) harder to classify, thoughts on this point is required.
 3. Choice of "t" seemed adhoc, incorporation of how "t" impacted the performance of the model would be interesting to see.

### Other comments
 * Section 4.1: It was not clear to me how exactly the "bad nodes" were introduced to the model. A little more details would be helpful for readers.

---

### Official Review · AnonReviewer1 · 2020-10-30
**limited novelty**

**Rating:** 4
**Confidence:** 5

**Review:**

This manuscript proposes FASG, a self-training model with GCN to improve node classification in graph. FASG introduces a checking part to add new nodes as supervision to enhance classification model. Experiments on several datasets show that FASG is better than some baseline methods.

Pros
1. The problem is important.
2. The presentation is good.
3. Introduce a new method to generate nodes with pseudo-labels.

Cons
1. The novelty of this work limited. According to my understanding, the contribution lies in the checking part and the checking part just uses GCN and SVM for generating pseudo-labeled nodes. In my opinion, it is a general choice and the novelty is limited.

2. The improvement over baseline methods is not significant. Most improvement percentages are smaller than 1% especially in Cora and PubMed datasets.

3. Experiments should be improved. Baseline methods are relatively weak, there are many work [1,2,3] for graph pre-training or self-training that could be compared and discussed.

[1] Strategies for Pre-training Graph Neural Networks, ICLR 2020

[2] GCC: Graph Contrastive Coding for Graph Neural Network Pre-Training, KDD 2020

[3] Gpt-gnn: Generative pre-training of graph neural networks, KDD 2020

---

### Decision · Program_Chairs · 2021-01-07
**Final Decision**

**Decision:**

Reject

**Comment:**

This paper presents a self-training idea for GCN models to help improve the node classification. The reviewers agreed that the technical contribution of the proposed approach is limited and the performance improvement seems marginal.